# The Impacts of Leaders’ Influence Tactics on Teleworkers’ Job Stress and Performance: The Moderating Role of Organizational Support in COVID-19

**DOI:** 10.3390/bs13100835

**Published:** 2023-10-12

**Authors:** Gukdo Byun, Jihyeon Rhie, Soojin Lee, Ye Dai

**Affiliations:** 1School of Business, Chungbuk National University, Cheongju 28644, Republic of Korea; bgukdo@cbnu.ac.kr (G.B.); ann2729g@cbnu.ac.kr (J.R.); 2College of Business Administration, Chonnam National University, Gwangju 61186, Republic of Korea; 3College of Business and Analytics, Southern Illinois University, Carbondale, IL 62901, USA; ye.dai@business.siu.edu

**Keywords:** influence tactics, job stress, perceived organizational support, job performance, COVID-19

## Abstract

With the outbreak of COVID-19, organizations have increased non-face-to-face work. This study aims to examine how leaders’ influence tactics affect employees’ psychological state and job performance in a non-face-to-face work (telework) setting. Moreover, based on substitutes for leadership theory, the study proposes that teleworkers’ perceived organizational support moderates the relationship between leaders’ influence tactics and their job stress in telework settings. We collected data via time-lagged surveys among 208 full-time employees in South Korean organizations that began teleworking after the outbreak of the COVID-19 pandemic. The results showed that leaders’ soft tactics (i.e., behaviors used to elicit the followers’ voluntary acceptance of a request) and rational tactics (i.e., behaviors that exert influence by providing empirical evidence based on reason or logic) significantly reduced teleworkers’ job stress, which in turn lowered their turnover intention and increased their task performance. Moreover, these tactics and teleworkers’ perceived organizational support interact to influence the workers’ job stress. By examining how leaders’ influence tactics affect teleworkers’ psychological stress, task performance, and turnover intention in the wake of the COVID-19 pandemic, this study theoretically broadens the influence tactics literature, which previously focused primarily on face-to-face workers. The study concludes with a discussion about the implications of findings and limitations, along with areas for future research.

## 1. Introduction

Following the COVID-19 pandemic in 2020, a rapid shift toward non-face-to-face methods of engagement across society occurred [1,2]. Organizations began to implement telework arrangements, such as telecommuting and remote and flexible work, which allowed employees to work from home rather than in an office. Particularly, telework refers to the practice of using advanced technologies to perform one’s work in a place other than one’s workplace instead of conducting production activities for one’s work only in a fixed place such as an office [3].

Previous research has evaluated the impact of telework on employee performance and psychological state. Some studies suggested that telework can help employees strike a work–life balance and increase productivity [4,5]; however, others reported the negative effects of telework on teleworkers’ emotional health [6,7]. Some researchers maintained that the sudden shift to telework may lead to psychological maladaptation issues, including isolation, worry, and anxiety, especially among workers accustomed to traditional office-based work [8]. Telework can be adopted as a new work arrangement in the post-pandemic world and serves as a major turning point in changes in the work environment [9,10,11]. Accordingly, it is critical to identify and examine practical ways to respond to and manage such changes effectively. Therefore, we focus on the role of leaders to improve teleworkers’ emotional well-being and retain and effectively manage them to increase their performance.

Leaders need to influence employees to achieve the organization’s goals. Accordingly, several researchers have studied ‘influence tactics’, which are defined as types of behaviors chosen by leaders to exert their influence effectively, over the past 40 years [11,12]. Specifically, there exist 11 categories of such tactics, including ingratiation, personal appeals, inspirational appeals, consultation, collaboration, rational persuasion, apprising, exchange, coalition, legitimating, and pressure [12]. Despite the possibility that these downward influence tactics may have a significant impact on achieving desirable attitudes and behaviors of their employees, little research has been conducted on this topic. Among a few existing ones, they were conducted in Western cultural settings [12,13]. Furthermore, prior research examined the effects of leaders’ influence tactics on workers in face-to-face work settings [14] but did not provide ready answers to whether similar tactics work in non-face-to-face settings. Thus, this current study seeks to understand in telework settings how leaders’ influence tactics affect workers’ job stress, turnover intention, and task performance in South Korea, a typical East Asian culture. Specifically, we aim to demonstrate how different categories of leaders’ influence tactics affect teleworkers’ psychological stress and how it influences their decision to stay with the organization, as well as their task performance at work.

Furthermore, this study also investigates how organizations can compensate for the influence of leaders on teleworkers. Perceived organizational support (POS) could be a crucial factor in evaluating the effect of leaders’ influence tactics on workers’ psychological stress. POS, proposed by Eisenberger et al. [15], is defined as “employees’ perception concerning the extent to which the organization values their contribution and cares about their well-being” (p. 51). They argued that a high level of POS evokes beneficial attitudes and behaviors among workers by increasing their emotional commitment to the organization and developing a sense of responsibility to reciprocate that commitment [16]. Accordingly, POS could act as a substitute for leadership in that provided support could forge or influence teleworkers’ perception of how much support they receive from the organization beyond that provided by the leader. Thus, based on substitutes for leadership theory [17], we examine how each of the influence tactics employed by leaders under the contingency of POS affects the psychological stress of teleworkers.

The purposes of this research are as follows. First, this study seeks to demonstrate the effect of leaders’ influence tactics on teleworkers’ workplace outcomes. Based on the meta-categories of influence tactics [12], this study investigates the individual effects of leaders’ hard, soft, and rational tactics on teleworkers’ psychological stress in telework situations. Furthermore, we examine whether teleworkers’ psychological stress mediates the relationship between leaders’ influence tactics and their turnover intention and task performance. Second, this study aims to show the moderating effect of teleworkers’ POS on the relationship between leaders’ influence tactics and teleworkers’ job stress. As leaders’ influence and workers’ POS serve as important metrics in determining individual workers’ psychological states, examining the possible substitution effects between these two variables allows us to develop a more nuanced understanding of the effect of each of these variables and their boundary conditions.

This study expands the applicable realm of influence tactics, the past study of which has concentrated mainly on face-to-face workers. First, it identifies how, amid the growing adoption of non-face-to-face telework settings, leaders’ influence tactics affect teleworkers’ psychological stress, turnover intention, and task performance. Second, the current study demonstrates the effectiveness of influence tactics among teleworkers in South Korea, a typical East Asian culture, and thus complements influence tactics research conducted in Western countries.

## 2. Hypothesis Development

### 2.1. Leaders’ Influence Tactics and Teleworkers’ Job Stress

Influence is the key to leadership [18]. As defined above, influence tactics refer to the specific types of behaviors that leaders choose to use to effectively exert influence over workers [12]. Several researchers have grouped influence tactics into meta-categories, the most widely used among them being hard, soft, and rational tactics [19,20]. Hard tactics employ an agent’s (leader’s) coercive power, such as reward and punishment, to directly or indirectly control the target (employee) [21]. Using soft tactics, the agent has no control over the target’s compliance [19]. The agent makes the requested goal more attractive to the target, encourages the target to follow the request voluntarily, and ensures that the target benefits from accepting the request [21]. Rational tactics are used by agents to exert influence by appealing to the target’s rationality based on reason or logic, such as regulations, laws, and codes [21]. We examine how these three meta-categories of influence tactics used by leaders (i.e., hard, soft, and rational tactics) affect teleworkers’ psychological stress.

First, the current study suggests that leaders’ hard tactics are positively correlated with teleworkers’ job stress. Telework puts workers in a different setting than face-to-face work. When workers were compelled to work virtually, they experienced unpleasant emotional and physiological states owing to a decreased level of control, which often led to increased stress [22]. Specifically, if leaders make coercive demands, threats, or exert a strong influence on teleworkers via formal authority in these settings, it will put teleworkers in a state of psychological anxiety due to the unfavorable work environment in which they have no control. This will likely cause stress [23,24]. Settings such as video conferencing, which are often used in telework contexts, may cause physical and mental fatigue owing to long meeting hours and difficulties in virtual interactions [25]. If leaders deliver messages in a coercive manner or make frequent demands and threats, teleworkers may spend more energy interacting with their leaders, which may inflict greater stress on them [26].

Some clues can be found in the literature about the positive correlation between leaders’ hard tactics and teleworkers’ stress. For example, Tepper [27] suggested that leaders’ coercive behaviors toward their subordinates may increase subordinates’ emotional stress. A meta-analysis by Zhang and colleagues [28] demonstrated that leaders’ negative influence on subordinates decreased organizational citizenship behaviors and increased counterproductive work behaviors via the subordinates’ psychological stress. In telework settings, where interaction requires more resources and energy, leaders’ hard tactics are likely to make workers experience psychological stress by reducing their sense of control and causing negative emotions. Therefore, we propose the following hypothesis:

**Hypothesis** **1-1:***Leaders’ hard tactics are positively associated with teleworkers’ job stress*.

Second, leaders’ soft tactics are expected to reduce teleworkers’ job stress. Soft tactics allow the target to voluntarily comply with or benefit from accepting a request [21]. When leaders use soft tactics, such as praising their subordinates or consulting with them directly, teleworkers are likely to build greater intimacy and reduce the sense of alienation caused by insufficient personal interaction in telework settings [18]. This is expected to decrease the level of psychological stress felt by their subordinates. Several researchers have highlighted the effectiveness of soft tactics. Barry and Shapiro [29] noted that soft tactics can be effective in helping secure a high level of compliance from subordinates. Also, Falbe and Yukl [30] claimed that soft tactics are most effective in eliciting voluntary participation from subordinates via consultation and inspirational appeals. Tepper and colleagues [31] proposed that leaders’ soft tactics may reduce workers’ resistance to assigned tasks and organizational change. Furthermore, the negative effect of hard tactics on resistance is also mitigated when they are used with soft influence tactics. Owing to a lack of interaction with coworkers and leaders, workers are more likely to suffer from different forms of stress in telework settings, including social isolation, presenteeism, difficulty using advanced technologies, anxiety regarding appraisal and reward, insecurity about employment continuity, and role ambiguity [23,24,25,26,32]. In these settings, leaders’ soft tactics may reduce their subordinates’ psychological stress, as these tactics help subordinates build greater intimacy with their supervisor and feel less alienated and anxious. Therefore, we propose the following hypothesis:

**Hypothesis** **1-2:***Leaders’ soft tactics are negatively associated with teleworkers’ job stress*.

Third, this study hypothesizes that leaders’ rational tactics will have a negative correlation with teleworkers’ job stress. Rational tactics exert influence by appealing to workers’ rationality, such as providing reasons or logic [21]. With the risk of a lack of interaction and clear instructions in telework settings, tactics such as presenting leaders’ goals, explaining the reasons for actions, and clarifying processes can reduce teleworkers’ stress by making them feel more comfortable in a predictable decision-making process. A communication process providing teleworkers with adequate information and feedback allows them to feel greater security and control, which then reduces their psychological stress. Some studies have provided relevant evidence. For instance, Tepper et al. [31] suggested that leaders who employ rational tactics are more effective in resolving conflicts with subordinates and increasing their job satisfaction. Falbe and Yukl [30] indicated that rational persuasion is more positively correlated with subordinates’ goal achievement and emotional commitment than pressure. Studies have shown that leaders who employ rational tactics are successful in achieving task goals, resolving conflicts, increasing subordinates’ job satisfaction, and improving employees’ task performance [33]. Therefore, we propose the following hypothesis:

**Hypothesis** **1-3:***Leaders’ rational tactics are negatively associated with teleworkers’ job stress*.

### 2.2. Teleworkers’ Job Stress and Turnover Intention

Stress is an unpleasant emotional and physiological state arising from unfavorable work experiences, such as uncertainty or outcomes beyond workers’ control [24]. When an organization is unable to cope effectively with stress-inducing settings and factors, it puts the organization’s members in a state of psychological anxiety or tension, which is called “job stress” [23]. Several studies have suggested that job stress leads to an increase in absenteeism and medical compensation and a decrease in productivity and employee performance [34,35,36,37]. Job stress may also lead to decreased job satisfaction [37], which may lead to increased turnover intention [38,39]. These negative consequences are not specific to a single industry or individual but are considered common across industries and individuals. Song and Gao [8] reported that teleworkers experience more stress than office-based workers. De Vries et al. [40] explained that telework increases public servants’ professional isolation and decreases their job commitment. It can be inferred from these studies that an increase in teleworkers’ job stress will increase their turnover intention.

The hypotheses framed thus far suggest that hard tactics are positively correlated with teleworkers’ job stress, and soft and rational tactics are negatively correlated with teleworkers’ job stress. We predict a positive relationship between teleworkers’ job stress and turnover intention based on previous studies. Therefore, job stress may mediate the relationship between leaders’ different forms of influence tactics and teleworkers’ turnover intention. Specifically, leaders’ hard tactics will increase teleworkers’ job stress, which in turn increases their intention to quit. Leaders’ soft and rational tactics will decrease workers’ job stress, which will also decrease their intention to quit. Thus, we propose the following hypotheses:

**Hypothesis** **2-1:***Teleworkers’ job stress mediates the positive association between leaders’ hard tactics and teleworkers’ turnover intention*.

**Hypothesis** **2-2:***Teleworkers’ job stress mediates the negative association between leaders’ soft tactics and teleworkers’ turnover intention*.

**Hypothesis** **2-3:***Teleworkers’ job stress mediates the negative association between leaders’ rational tactics and teleworkers’ turnover intention*.

### 2.3. Teleworkers’ Job Stress and Task Performance

Job stress, with underlying factors such as unfavorable and threatening work environments and interpersonal and role conflicts, undermines the psychological well-being of workers [23]. This, in turn, results in reduced productivity and performance, as indicated in many previous studies [34,35,36,41,42]. According to Margolis and Kroes [42], workplace stress can have a detrimental influence on workers’ productivity. In addition, Khuong and Yen [43] found that even among business leaders, elevated stress has a very significant negative influence on their performance, which then reduces organizational performance. They argued that business leaders with a high level of stress may choose an unproductive coping method to relieve their stress, which may lead to negative performance. Similarly, teleworkers’ job stress is likely to hinder their ability to finish tasks. Thus, this study hypothesizes that leaders’ hard tactics increase teleworkers’ job stress, which then decreases their task performance, whereas leaders’ soft and rational tactics decrease teleworkers’ job stress, which then increases their task performance. Therefore, we propose the following hypotheses:

**Hypothesis** **3-1:***Teleworkers’ job stress mediates the negative association between leaders’ hard tactics and teleworkers’ task performance*.

**Hypothesis** **3-2:***Teleworkers’ job stress mediates the positive association between leaders’ soft tactics and teleworkers’ task performance*.

**Hypothesis** **3-3:***Teleworkers’ job stress mediates the positive association between leaders’ rational tactics and teleworkers’ task performance*.

### 2.4. The Moderating Effect of POS

Perceived organizational support (POS) relates to employees’ perceptions of how much their efforts and contributions are appreciated by their organization [44]. From the perspective of social exchange theory [45], organizational support theorists perceive employment as an exchange of employees’ efforts and loyalty for material rewards and social resources provided by the organization [16,46]. In addition, employees are posited to develop a common perception of how much the organization values their contributions and welfare, which shapes their behaviors and attitudes toward the organization [46]. Studies have shown that employees’ POS may not only exert a direct effect on their performance but also serve as a key contingency factor for employee performance. For example, Erdogan and Enders [47] verified the moderating role of supervisors’ POS on the relationships between leader–member exchange (LMX), job satisfaction, and job performance. In addition, Malik and Noreen [48] demonstrated that POS moderates the link between occupational stress and affective well-being among teachers. Following this line of reasoning, this study intends to investigate the moderating role of POS on the effect of leaders’ influence tactics on teleworkers’ job stress.

Substitutes for leadership theory [17] have identified factors that can substitute leaders’ behaviors during the process when leadership is exercised and have further stated that such factors can neutralize or weaken leaders’ influence. Specifically, the theory proposes that subordinate, task, and organizational contingency factors can all serve as substituting factors that can negate, neutralize, or facilitate the effects of leadership [17,49]. Substitutes make the practice of leadership unnecessary, and thus, the practice of leadership is not expected to produce the same effect as organizational, task, or subordinate contingency factors [17]. For instance, formalized organizational structures and structured tasks serve as substitutes for leaders’ structure-oriented behaviors. Certain situations that do not require a relationship with leaders to satisfy the needs of workers act as substitutes for leadership. Thus, POS may serve as a key contingency factor that can substitute the effect of leadership by representing workers’ perceptions of how much additional assistance and guidance they can receive from the organization beyond those provided by the leader.

The positive association between leaders’ hard tactics and teleworkers’ job stress will be stronger when POS is lower. This study proposes that the employment of leaders’ hard tactics—comprising only threats, punishments, and one-way notices without providing support or care to socially isolated teleworkers—may increase their stress. In such situations, high POS may act as a major substitute for leadership [17]. Workers will develop the perception that they are supported by the organization, which is expected to negate the effect of leaders’ hard tactics. If POS is low, it is not expected to act as a substitute and will strengthen the detrimental impact of leaders’ hard tactics on job stress. This study thus hypothesizes that POS will moderate the relationship between leaders’ hard tactics and teleworkers’ job stress.

**Hypothesis** **4-1:***Teleworkers’ POS moderates the association between leaders’ hard tactics and teleworkers’ job stress such that the positive association between leaders’ hard tactics and teleworkers’ job stress is stronger when POS is lower than when it is higher*.

Next, this study predicts that the negative association between leaders’ soft tactics and teleworkers’ job stress will be stronger when POS is lower. Rhoades and Eisenberger [50] claimed that higher POS leads to workers’ increased loyalty and performance, as workers believe that rewards and evaluations are fair in their organization. Thus, a high level of POS may act as a major substitute for leadership without leaders’ influence, as teleworkers perceive that their organization recognizes and supports their contributions [17]. A high level of POS means that teleworkers believe that their contributions are properly recognized by their organization and that they are evaluated and rewarded accordingly. This will substitute the effect of soft tactics, including personal attention and the warm influence of leaders. As a result, the effectiveness of leaders’ soft tactics on teleworkers’ psychological stress may be weakened because leaders’ influence could be substituted by a high level of organizational support perceived by teleworkers. However, a low level of POS cannot provide adequate substitution. Thus, leaders’ influence on teleworkers may loom more prominent when their perception of organizational support is low because nothing can replace the influence or role of a leader. Therefore, this low level of POS is expected to strengthen the effect of soft tactics when leaders make efforts to increase teleworkers’ felt personal intimacy and decrease their sense of alienation. Thus, we propose the following hypothesis:

**Hypothesis** **4-2:***Teleworkers’ POS moderates the association between leaders’ soft tactics and teleworkers’ job stress such that the negative association between leaders’ soft tactics and teleworkers’ job stress is stronger when POS is lower than when it is higher*.

Finally, this study predicts that the negative association between leaders’ rational tactics and teleworkers’ job stress will be stronger when POS is lower. The study hypothesizes that high POS—in which teleworkers perceive that their organization gives them credit, recognizes their contributions, and provides them with psychological and material support—will serve as a strong substitute for leaders’ influence [17]. Thus, high POS is expected to make it difficult to produce the positive impact of rational tactics, in which leaders exert influence via rationality based on facts and logic. In other words, due to teleworkers’ high level of perceived organizational support, which can at least partially substitute leaders’ influence, the impact of leaders’ soft tactics on their psychological stress may be reduced. However, workers’ low levels of POS may not adequately substitute leaders’ roles in reducing teleworkers’ stress and may not work as effectively as leaders’ reliance on rational tactics to reduce teleworkers’ elevated anxiety about evaluations and promotions. That is, when their perception of organizational support is low since nothing can replace a leader’s role, the effect of leaders’ rational tactics on teleworkers will loom conspicuous. Therefore, the effect of leaders’ rational tactics will be weakened by a high POS, which can act as a substitute for leadership, and the negative association between leaders’ rational tactics and teleworkers’ job stress would be stronger when POS is lower. We thus propose the following hypothesis:

**Hypothesis** **4-3:***Teleworkers’ POS moderates the association between leaders’ rational tactics and teleworkers’ job stress such that the negative association between leaders’ rational tactics and teleworkers’ job stress is stronger when POS is lower than when it is higher*.

Figure 1 shows our research model.

## 3. Method

### 3.1. Sample and Data Collection

This study collected data using time-lagged(T1/T2) surveys that were randomly distributed among full-time employees in organizations that began to adopt telework because of the COVID-19 pandemic in South Korea. These organizations were chosen based on the network of the research team, and data were obtained via convenience sampling [51]. With the support of each organization’s HR manager, we conducted not only offline surveys but also online surveys in places where offline surveys were not available. We obtained IRB approvals from the Research Ethics Review Committee of Chungbuk National University. The questionnaire was coded in such a way that anonymity was guaranteed, and we distributed them with a cover letter, ensuring confidentiality and that the data would only be utilized for academic research purposes. For the online survey, we sent an invitation email with the same cover letter to employees by including a trackable link to the Time 1 questionnaire. For the first survey (T1), 500 copies of the survey were sent to those who volunteered to participate in our research, both online and offline, and 368 of them were returned, yielding a response rate of 73.6%. A total of 337 completed surveys (67.4%) were obtained after excluding copies with missing and incomplete responses. Two months after the first survey, the second survey (T2) was conducted. Questionnaires were distributed to 337 respondents, of which 242 were returned. Of these, 208 copies (final response rate: 41.6%) were used for the final analysis after removing 34 surveys that could not be used (such as those that could not be matched to the first questionnaire or those that had incomplete responses). Given the sample to the number of variables ratio, the sample size is considered sufficient [51,52]. The sample had the following characteristics: 121 respondents (58.2%) were men, and 87 respondents (41.8%) were women. In terms of age, 59 respondents (28.4%) were in their 20s, and 62 respondents (29.8%) were in their 30s, followed by 43 respondents (20.7%) who were in their 40s, 31 respondents (14.9%) in their 50s, and 13 respondents (6.3%) in their 60s and above. Furthermore, 136 respondents (65.4%) had a 4-year university degree, followed by 50 respondents (24%) who had graduate school degrees. Respondents worked in the service (51, 24.5%), manufacturing (41, 19.7%), and education/counseling (26, 12.5%) industries.

### 3.2. Measurements

This study measured independent variables of leader influence tactics, a moderating variable of POS, control variables of demographic data including gender, age, education, industry, and tenure with supervisor in the first survey, a mediating variable of job stress in the second survey, and dependent variables of turnover intention and task performance in the second survey. Thus, we collected data at two different time points to reduce the likelihood of reversed causality and common method biases [53]. Seven-point Likert-type scales were used for all questions except for those on demographic characteristics ranging from 1 (strongly disagree) to 7 (strongly agree).

*Influence tactics.* We use a total of 44 items developed by Yukl and colleagues [12] to measure influence tactics. Specifically, 16 questions were used for hard tactics, with four for each of legitimating, pressure, coalition, and exchange tactics. Examples of the questions included the following: “My boss demands that I carry out a request” and “My boss repeatedly checks to see if I have carried out a request”. A total of 20 questions were used for soft tactics, with four for each of ingratiation, consultation, collaboration, inspirational appeal, and personal appeals. Examples of the questions included the following: “My boss praises my past performance or achievements when asking me to do a task for him/her” and “My boss talks about ideals and values when proposing a new activity or change”. A total of eight questions were used for rational tactics, with four for each of rational persuasion and apprising. Examples of the questions included the following: “My boss explains clearly why a request or proposed change is necessary to attain a task objective”, and “My boss describes benefits I could gain from doing a task or activity (e.g., learn new skills, meet important people, enhance my reputation)”.

*Job stress.* Job stress was measured using Kim’s [54] 4-item scale, which was developed to measure workers’ physical and mental reactions to problems when carrying out their job or the work environment. The following are examples of the questions included in the survey: “I often feel anxious because of my work”, “I often feel depressed because of my work”, and “I often feel frustrated because of my work”.

*POS*. POS was measured using seven items that Eisenberger et al. [15] developed to measure workers’ perceptions of how much their contributions were recognized and appreciated by their organization. The following are examples of the questions included in the survey: “My organization values my contributions to its well-being”, “My organization is willing to help me when I need a special favor”, and “My organization strongly considers my goals and values”.

*Turnover intention.* To measure turnover intention, three items [55] were used. The following are examples of the survey questions: “I often think of leaving the organization” and “It is very possible that I will look for a new job next year”.

*Task performance.* This study used a 7-item scale from Williams and Anderson [56] to measure the extent to which teleworkers perform the role formally required by their organization. The following are examples of the survey questions: “I meet formal performance requirements of the job” and “I fulfill responsibilities specified in the job description”.

*Control variables.* The control variables included gender, age, education, industry, and tenure with the supervisor.

### 3.3. Analytic Strategy

To investigate our hypotheses, we conducted hierarchical multiple regression analyses. We mean-centered the variables before creating the interaction terms [57] to address potential multicollinearity issues. Additionally, we tested mediation effects by using a bootstrapping method [58].

## 4. Results

Table 1 shows the descriptive statistics for the variables in the model and the correlation coefficients among the variables. Cronbach’s Alpha, which indicates the reliability of a variable, was ≥0.79 for all variables except for turnover intention (a = 0.61). This confirmed that there was high internal consistency. For turnover intention, a combination of forward and reverse questions was used in the questionnaire, which may have led to a problem with reliability [59]. However, there was no problem when only forward questions were used, as Cronbach’s Alpha was 0.80 [60], indicating high levels of reliability. The correlation coefficients between variables were similar to those in previous studies, and no correlation coefficient was above 0.80, confirming that there was no issue with multicollinearity. To ensure the validity of the measures, we performed both exploratory factor analysis (EFA) and confirmatory factor analysis (CFA) on all variables of this study and confirmed that each measure of all selected variables was valid and at least acceptable.

Table 2 presents the results of hierarchical regression analyses. Hypothesis 1-1, which suggested a positive association between leaders’ hard tactics and teleworkers’ job stress, was not supported (*β* = −0.05, ns, in Model 2, Table 2). Hypothesis 1-2, which proposed a negative association between leaders’ soft tactics and teleworkers’ job stress, was supported (*β* = −0.16, *p* < 0.05, Model 3). Hypothesis 1-3, which suggested a negative association between leaders’ rational tactics and teleworkers’ job stress, was supported (*β* = −0.21, *p* < 0.01, Model 4).

Next, this study hypothesized that teleworkers’ job stress would mediate the relationship between leaders’ influence tactics and teleworkers’ turnover intention and task performance. The study conducted a bootstrapping analysis, as suggested by Preacher et al. [58], to test the hypotheses. Bootstrapping was conducted with 10,000 replicates at the 90% confidence interval. Table 3 presents the results. Hypothesis 2-1, which predicted that teleworkers’ job stress would mediate the association between leaders’ hard tactics and teleworkers’ turnover intention, was not significant as the 90% confidence interval included zero (LL 90% CI = −0.068, UL 90% CI = 0.022). Therefore, Hypothesis 2-1 was not supported. Hypothesis 2-2, which predicted that teleworkers’ job stress would mediate the association between leaders’ soft tactics and teleworkers’ turnover intention, had a significant mediating effect as the 90% confidence interval did not include zero (LL 90% CI = −0.104, UL 90% CI = −0.016). Therefore, Hypothesis 2-2 was supported. Hypothesis 2-3, which predicted that teleworkers’ job stress would mediate the association between leaders’ rational tactics and teleworkers’ turnover intention, had a significant mediating effect as the 90% confidence interval did not include zero (LL 90% CI = −0.080, UL 90% CI = −0.019). Therefore, Hypothesis 2-3 was supported.

Bootstrapping analysis was conducted to test the hypotheses of the mediating effect of teleworkers’ job stress on the association between leaders’ influence tactics and teleworkers’ task performance. Table 4 presents the results. Hypothesis 3-1, which predicted that teleworkers’ job stress would mediate the association between leaders’ hard tactics and teleworkers’ task performance, was not significant as the 90% confidence interval included zero (LL 90% CI = −0.004, UL 90% CI = 0.026). Therefore, Hypothesis 3-1 was not supported. Hypothesis 3-2, which predicted that teleworkers’ job stress would mediate the association between leaders’ soft tactics and teleworkers’ task performance, had a significant mediating effect as the 90% confidence interval did not include zero (LL 90% CI = 0.001, UL 90% CI = 0.047). Therefore, Hypothesis 3-2 was supported. Hypothesis 3-3, which predicted that teleworkers’ job stress would mediate the association between leaders’ rational tactics and teleworkers’ task performance, had a significant mediating effect as the 90% confidence interval did not include zero (LL 90% CI = 0.002, UL 90% CI = 0.038). Therefore, Hypothesis 3-3 was supported.

A hierarchical regression analysis was performed to examine whether POS moderates the association between leaders’ influence tactics and teleworkers’ job stress. Table 2 shows the results. To test the moderating effect, this study investigated whether the interaction term between the independent and moderating variables had a statistically significant impact on the dependent variable. We mean-centered the independent variables and moderator to mitigate a potential multicollinearity issue [57]. To confirm the specific direction of the moderating effect, we plotted the findings using the method given by Aiken et al. [57].

Hypothesis 4-1, which predicted that teleworkers’ POS moderates the association between leaders’ hard tactics and teleworkers’ job stress, was not supported (*β* = 0.02, ns, Model 6, Table 2). However, Hypothesis 4-2, which proposed that teleworkers’ POS would moderate the association between leaders’ soft tactics and teleworkers’ job stress, was supported (*β* = 0.14, *p* < 0.05, Model 8). We plotted the results in Figure 2 [57]. As seen in the figure, the negative association between leaders’ soft tactics and teleworkers’ job stress was stronger when POS was lower.

Hypothesis 4-3, which predicted that teleworkers’ POS would moderate the association between leaders’ rational tactics and teleworkers’ job stress, was supported (*β* = 0.15, *p* < 0.05, Model 10). This interaction effect is shown in Figure 3. As predicted, the negative association between leaders’ rational tactics and teleworkers’ job stress was stronger when POS was lower.

## 5. Discussion

This study examined the effect of leaders’ influence tactics on teleworkers’ job stress, turnover intention, task performance, and the moderating effect of POS among full-time employees who experienced a new telework arrangement because of the COVID-19 epidemic. The study found that leaders’ soft and rational tactics reduce teleworkers’ job stress, which in turn lowers their turnover intention and increases their task performance. In addition, there was an interaction between these tactics and teleworkers’ POS in influencing job stress. These results lead to the following theoretical and practical contributions.

### 5.1. Theoretical Implications

First, this study demonstrated the effect of leaders’ influence tactics (i.e., hard, soft, and rational tactics) on teleworkers’ job stress. Previous studies have shown that job stress increases workers’ negative organizational behaviors, including absenteeism, turnover, and job withdrawal behavior while having a detrimental impact on job satisfaction, organizational commitment, and productivity [34,35,36,61]. This study empirically demonstrated that soft and rational tactics are more effective than hard tactics, such as pressure, legitimating, and coalition tactics, in reducing teleworkers’ stress. In fact, some researchers have suggested that it may not be sufficient to lead in virtual situations using only recognized social skills, such as the features of good face-to-face communication [62,63]. Accordingly, this study fills a gap in the literature by demonstrating the effectiveness of individual influence tactics in telework situations based on meta-categories, which have not been clearly addressed in previous studies, thus expanding the tactics’ applicability in practice. It confirmed that leaders’ influence tactics are applicable to teleworkers.

It is also noteworthy that we cannot find support for the association between leaders’ hard tactics and teleworkers’ job stress. Teleworkers communicate with their leaders using a variety of methods, such as text messages and phone calls in non-face-to-face working situations. It is even more difficult to read others’ non-verbal messages in a video conference than in face-to-face situations due to online connectivity issues and image quality issues [26,64]. It is speculated that the non-finding could be because leaders’ coercive behavior and language seem less threatening in non-face-to-face settings and can be ignored more easily by teleworkers to avoid experiencing stress. Moreover, our findings are consistent with those found by Moideenkutty and Schmitdt [65], which indicate that while leaders’ positive influence tactics had a significant effect on organizational citizenship behavior, the relationship between leaders’ negative influence tactics and organizational citizenship behavior was not significant. The authors pointed out that this result could be due to the high power distance in Islamic national cultures, where employees are more receptive to leaders’ negative influence tactics. The same factor could also work in South Korea, a country with high power distance [66]. Thus, future studies could explicitly include power distance and other cultural dimensions as influencing factors when examining the effect of influence tactics.

Second, this study demonstrated the mediating effect of teleworkers’ job stress on the association between leaders’ influence tactics and teleworkers’ turnover intention and task performance. Not only does this improve our understanding of how influence tactics work but it also provides a concrete pathway for leaders to make the case for the effectiveness of using influence tactics [12], which was identified as a limitation in prior studies. Further, this study highlighted the importance of stress management for employees and organizations by explaining how job stress is a major factor in the process, which can cause teleworkers to leave their organizations or undermine their task performance.

Third, this study confirmed that POS could serve as a contingency factor for the effect of leaders’ influence tactics on employee turnover intention and performance. Drawing from substitutes for leadership theory [17], this study found that leaders’ soft and rational tactics have a stronger effect on reducing teleworkers’ job stress when their POS is lower in telework settings. In other words, teleworkers require greater assistance from their organizations because they are more likely to experience social isolation and anxiety about performance evaluation. In such situations, increased POS can be used as a substitute for leadership when effective leadership is inadequate. In contrast, ineffective or limited organizational support may highlight the importance of the role of leaders. If leaders develop personal intimacy with teleworkers, praise them, and rationally assign them tasks, teleworkers may overcome negative consequences arising from the lack of organizational support and experience lower job stress. This study confirmed that this positive impact may improve teleworkers’ task performance and reduce their turnover intention.

### 5.2. Practical Implications

Our findings have the following practical implications. First, to reduce teleworkers turnover rate and increase task performance, leaders could resort to soft and rational influence tactics, such as inspirational appeals and rational persuasion, but restrain themselves from using hard tactics, such as pressure, legitimization, and association, that may lead to adverse outcomes or have limited effect in telework settings. Moreover, managers need to recognize that the psychological stress of teleworkers can significantly increase or decrease depending on what influence tactics the leader uses. Accordingly, this suggests that managers need to be more sensitive and considerate in their attempts to exert influence. Therefore, organizations need to place more emphasis on the effectiveness of these influence tactics in leadership training and encourage leaders to learn and use them in management practices.

Second, organizations need to be fully conscious of teleworkers’ well-being and be more attentive to providing support for them. Based on the results of this study, when teleworkers perceive that their organizations do not provide enough material and emotional support for them, they become more sensitive to their leaders’ behaviors and opinions. On the other hand, a high level of teleworkers’ perceived organizational support can mitigate their stress level to some extent, even when leaders do not exercise sufficient soft and rational tactics.

## 6. Limitations and Future Research

Nevertheless, this study has a few limitations. First, it might still be difficult to generalize our results based on data collected from only several industries as different types of non-face-to-face respondents have different work arrangements and periods. It is necessary for future researchers to expand the scope of their studies and verify the results of this study by including employees having different job titles from a wider range of industries. Second, common method bias, although mitigated, may still have exerted its influence on study results as the data were collected from the same source. Future research should overcome these limitations by using objective data on the influence of job stress and task performance or by separating the respondents into supervisors and subordinates.

## 7. Conclusions

Leaders can motivate or disengage employees via their influence tactics. This study provides implications about the types of influence tactics that leaders need to resort to in a period when more and more companies are increasing their share of non-face-to-face work arrangements. We hope that our study serves as a starting point based on which future researchers can further create a stronger theoretical foundation for the functioning of influence tactics and thus guide leaders’ behaviors in organizations in this new teleworking era.

## Figures and Tables

**Figure 1 behavsci-13-00835-f001:**
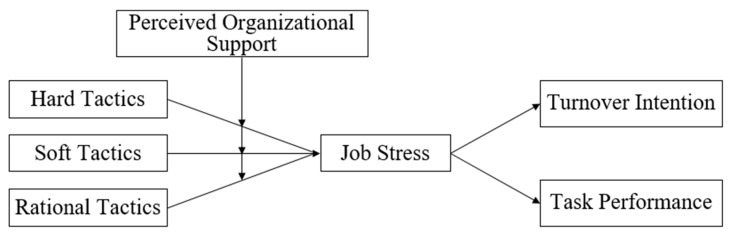
Hypothesized model.

**Figure 2 behavsci-13-00835-f002:**
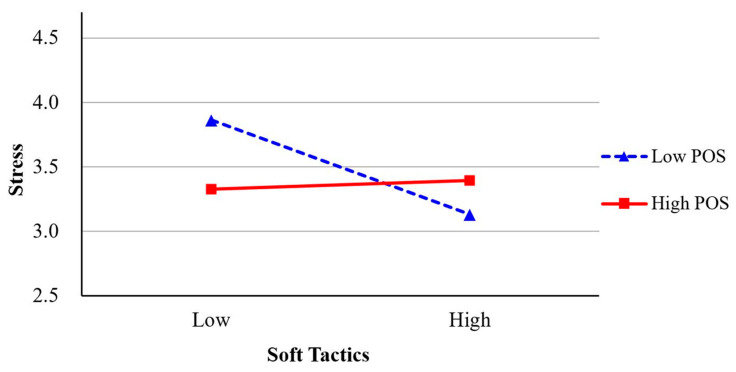
The moderating effect of POS on the relationship between soft tactics and stress.

**Figure 3 behavsci-13-00835-f003:**
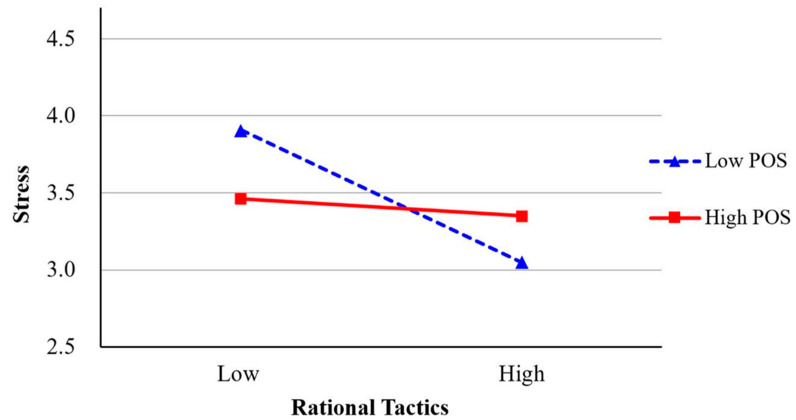
The moderating effect of POS on the relationship between rational tactics and stress.

**Table 1 behavsci-13-00835-t001:** Descriptive statistics and correlations.

Variable	M	SD	1	2	3	4	5	6	7	8	9	10	11	12
1. Gender	0.42	0.49												
2. Age	2.41	1.22	−0.45 **											
3. Education	3.13	0.67	−0.19 **	0.28 **										
4. Industry	5.57	3.02	0.28 **	−0.14 *	0.03									
5. Tenure with supervisor	20.50	19.02	−0.10	0.07	−0.05	−0.20 **								
6. Hard tactics	3.11	0.97	0.17 *	−0.17 *	−0.16 *	−0.01	0.13	(0.89)						
7. Soft tactics	3.90	1.44	0.32 **	−0.43 **	−0.22 **	0.07	0.11	0.55 **	(0.92)					
8. Rational tactics	3.88	1.03	0.39 **	−0.39 **	−0.22 **	0.14	−0.01	0.54 **	0.76 **	(0.95)				
9. Perceived organizational support	4.53	1.35	0.22 **	−0.23 **	−0.19 **	0.02	0.05	0.25 **	0.55 **	0.58 **	(0.91)			
10. Job stress	3.54	1.24	−0.16 *	0.02	−0.19 **	−0.14 *	−0.04	−0.05	−0.19 **	−0.16 *	−0.10	(0.79)		
11. Turnover intention	4.17	1.20	−0.16 *	0.10	−0.01	−0.12	0.01	−0.11	−0.21 **	−0.21 **	−0.41 **	0.31 **	(0.61)	
12. Task performance	5.20	0.81	−0.05	0.08	0.09	0.00	0.05	−0.15 *	0.02	−0.03	0.00	−0.12	−0.08	(0.78)

*Note:* N = 208. Reliability is placed along the diagonal in parentheses. * *p* < 0.05; ** *p* < 0.01 (two-tailed); 1. Gender: 0 = male, 1 = female; 2. Age: 1 = 20s, 2 = 30s, 3 = 40s, 4 = 50s, 5 = 60 and older; 3. Education: 1 = high school graduate, 2 = associate degree, 3 = bachelor’s degree, 4 = master’s degree and above, 5 = other; 4. Industry: 1 = manufacturing, 2 = finance/accounting, 3 = logistics, 4 = construction/electrical company, 5 = IT, 6 = service, 7 = medical/pharmaceutical, 8 = art/design/advertisement, 9 = education/consultation, 10 = patent, 11 = welfare, 12 = other; 5. Tenure with supervisor: number of months.

**Table 2 behavsci-13-00835-t002:** Regression results for hypotheses: main and moderating effects.

	Stress									
	M1	M2(H1-1)	M3(H1-2)	M4(H1-3)	M5	M6(H4-1)	M7	M8(H4-2)	M9	M10(H4-3)
1. Control variables										
Gender	−0.16 *	−0.18 *	−0.15	−0.15	−0.17 *	−0.17 *	−0.15	−0.15	−0.15	−0.16 *
Age	−0.01	−0.02	−0.05	−0.09	−0.03	−0.03	−0.05	−0.10	−0.09	−0.13
Education	−0.22	−0.23 *	−0.23	−0.24	0.24 **	−0.23 **	−0.24 **	−0.24 **	−0.24 **	−0.25 ***
Industry	−0.11	−0.11	−0.10	−0.10	−0.11	−0.11	−0.10	−0.11	−0.10	−0.12
Tenure with supervisor	−0.09	−0.09	−0.09	−0.06	0.08	−0.08	−0.08	−0.07	−0.06	−0.05
2. Main effect										
Hard tactics		−0.05			−0.03	−0.03				
Soft tactics			−0.16 *				−0.14	−0.13		
Rational tactics				−0.21 **					−0.21 *	−0.20 *
POS *					−0.10	−0.10	−0.03	−0.06	−0.01	−0.03
3. Moderating effect										
Hard tactics × POS						0.02				
Soft tactics × POS								0.14 *		
Rational tactics × POS										0.15 *
Overall F	3.95 **	3.36 **	4.08 **	4.75 ***	3.15 **	2.76 **	3.51 **	3.65 **	4.05 ***	4.18 ***
R^2^	0.09	0.09	0.11	0.12	0.10	0.10	0.11	0.13	0.12	0.14
Changed in F		0.48	4.42 *	8.09 **	1.83	0.10	0.17	4.28 *	0.01	4.53 *
Changed in R		0.00	0.02	0.04	0.01	0.00	0.00	0.02	0.00	0.02

*Note:* * POS: Perceived organizational support; * *p* < 0.05; ** *p* < 0.01; *** *p* < 0.001 (two-tailed).

**Table 3 behavsci-13-00835-t003:** Indirect effect of influence tactics on the turnover intention of teleworkers.

Variables	Bootstrap Results for Indirect Effect			
	Effect	Boot SE	LL 90% CI	UL 90% CI
Hard tactics	−0.018	0.029	−0.068	0.022
Soft tactics	−0.052	0.030	−0.104	−0.016
Rational tactics	−0.050	0.021	−0.080	−0.019

*Note*: N = 208. Bootstrap sample size = 10,000. SE = standard error; LL = lower limit; CI = confidence interval; UL = upper limit.

**Table 4 behavsci-13-00835-t004:** Indirect effect of influence tactics on the task performance of teleworkers.

Variables	Bootstrap Results for Indirect Effect			
	Effect	Boot SE	LL 90% CI	UL 90% CI
Hard tactics	−0.005	0.009	−0.004	0.026
Soft tactics	0.016	0.013	0.001	0.047
Rational tactics	0.014	0.011	0.002	0.038

*Note:* N = 208. Bootstrap sample size = 10,000. SE = standard error; LL = lower limit; CI = confidence interval; UL = upper limit.

## Data Availability

The data that support the findings of this study are available upon request from the corresponding author.

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
