# Peer review of "The Impacts of Leaders’ Influence Tactics on Teleworkers’ Job Stress and Performance: The Moderating Role of Organizational Support in COVID-19"

_behavsci, 2023, doi:10.3390/bs13100835_

Round 1
Reviewer 2 Report
Thank you for allowing me to review this paper. However, I have shared my viewpoints.
The title needs to be shorter to understand. Try to make it precise.
In the abstract, you must include a short methodology (sample size, sampling technique). Besides, you must include the implications, limitations and areas for future research. Besides, you have written POS; I understand it is perceived as organizational support. But, the readers might need to help understanding it. So, in the very first time, you should clear it.
The introduction part is well-written.
The literature review part needs to improve by adding theoretical support. Your variables (independent, dependent and moderating variables) must come from theory; based on those variables, you can draw your framework and formulate the hypotheses.
The methodology section is well structured. But I have some observations.
1. Why do you need to collect the data in two stages?
2. Which sampling techniques you used to collect the data?
3. Finally, you have selected 208 copies (final response rate: 61.17%) that were used for the final analysis. Give your justification that the sample size is adequate.
You can read the following article to get an idea regarding sampling techniques and sample size.
http://scientificia.com/index.php/JEBE/article/view/201
https://sciendo.com/article/10.2478/seeur-2022-0023
You can add the analysis plan and ethical issues in the methodology section.
In the result section, it is quite OK.
One of the main shortcomings of this article is the contribution part. The contribution must have at least two sections (1) Theoretical contributions and (2) Practical contributions.
Finally, you must include “Conclusion, limitations, and areas for future research."
Thanks.
Reviewer 3 Report
An interesting and promising study, but needs improvement.
Intro: Gives an indication into the importance of the study, but needs a clear problem statement. Why is this study important in the context of remote workers in South Korea? What issues are remote workers facing in this area that justifies this study?
Method: Major details are missing regarding sampling procedures - how do the authors define random sampling in this context, how was the paper survey administered, why physical survey and not online survey, how were participants selected, what was the sampling frame, why 500 copies (how was this sample size determined), was it IRB approved, how were the time frames between both surveys determined, were responses between both surveys matched (why or why not), etc. Details on the target population is also needed.
While survey questions came from existing measures, I would suggest conducting an exploratory factor analysis and confirmatory factor analysis to identify any items with low factor loadings within constructs as items can change within different contexts.
Discussion: This section mostly repeats the results. What insights can the authors suggest as to how leaders can potentially use these tactics with remote employees and within specified industries? How do results fit into the theoretical gap the study sought to address? In the context of this study, what does this mean for remote employees and their employers in South Korea? What are the practical implications of this study based on the results?
References: There are other recent studies that investigated remote employer-employee relationships, and remote employee organizational commitment that could also be appropriate here.
Good. A few minor errors that can be corrected with proofreading.
Round 2
Reviewer 2 Report
Dear authors
Thank you for the modification of your manuscripts.
However, best of luck.
Regards.
Reviewer 3 Report
Very good job in addressing this reviewer's comments. Just a few suggested edits to these sentences -
L. 46 - "Accordingly, it is critical to identify and examine practical ways to respond to and management such changes effectively"; change to manage
L. 515 - "Thus, Future researcher could explicitly include power distance and other cultural dimensions as influencing factors when examining the 516 effect of influence tactics"; suggested change to future studies
L. 541 - "First, to reduce employees'..."; indicate remote employees or say teleworkers
L. 552 - Same as L. 541; indicate it's in reference to remote employees or teleworkers
L. 553- 557 - "Based on the results of this study, when employees perceive that their organizations do not provide enough material and emotional support for them, they became more sensitive to their leaders' behaviors and opinions. On the other hand, a high level of employees' perceived organizational support can mitigate their stress level to some extent, even when leaders do not exercise sufficient soft and rational tactics." - Excellent insight!
Great work! Good luck with your future research projects.